# Conditional Controllable Image Fusion

**Bing Cao**[1,2]   **Xingxin Xu**[3]   **Pengfei Zhu**[1]*   **Qilong Wang**[1]   **Qinghua Hu**[1]

[1]College of Intelligence and Computing, Tianjin University, Tianjin, China
[2]State Key Laboratory of Integrated Services Networks, Xidian University, Xi'an, China
[3]School of New Media and Communication, Tianjin University, Tianjin, China
`{caobing, xuxingxin, zhupengfei, qlwang, huqinghua}@tju.edu.cn`

## Abstract

Image fusion aims to integrate complementary information from multiple input images acquired through various sources to synthesize a new fused image. Existing methods usually employ distinct constraint designs tailored to specific scenes, forming fixed fusion paradigms. However, this data-driven fusion approach is challenging to deploy in varying scenarios, especially in rapidly changing environments. To address this issue, we propose a conditional controllable fusion (CCF) framework for general image fusion tasks without specific training. Due to the dynamic differences of different samples, our CCF employs specific fusion constraints for each individual in practice. Given the powerful generative capabilities of the denoising diffusion model, we first inject the specific constraints into the pre-trained DDPM as adaptive fusion conditions. The appropriate conditions are dynamically selected to ensure the fusion process remains responsive to the specific requirements in each reverse diffusion stage. Thus, CCF enables conditionally calibrating the fused images step by step. Extensive experiments validate our effectiveness in general fusion tasks across diverse scenarios against the competing methods without additional training. The code is publicly available.[†]

## 1   Introduction

Image fusion aims at integrating complementary information from multi-source images, fusing a new composite image containing richer details [1]. It has been applied in various scenarios that single image contains incomplete information, such as multi-modal fusion (MMF) [2, 3], multi-exposure fusion (MEF) [4, 5], multi-focus fusion (MFF) [6], and remote sensing fusion [2]. The fused image inherits the strengths of both modalities, resulting in a composite with enhanced visual effects [7]. These fusion tasks have diverse downstream applications in computer vision, including object detection [8–10], semantic segmentation [11, 12], and medical diagnosis [13] because the comprehensive representation of images with multi-scene information contributes to the improved performance of applications.

Recently, numerous image fusion methods [14–17] have been proposed, such as traditional fusion methods [18], CNN-based fusion methods [19, 20] and GAN-based methods [21]. While these methods produce acceptable fused images in certain scenarios, they are also accompanied by significant drawbacks and limitations: (i) They are often tailored for specific scenarios or individual tasks, limiting their adaptability across diverse applications; (ii) These methods necessitate training and consume substantial computational resources, posing limitations in terms of time and resource requirements. Lately, denoising diffusion probabilistic models (DDPM) have emerged as an iterative

---

*Corresponding author.
[†]`https://github.com/jehovahxu/CCF`

38th Conference on Neural Information Processing Systems (NeurIPS 2024).

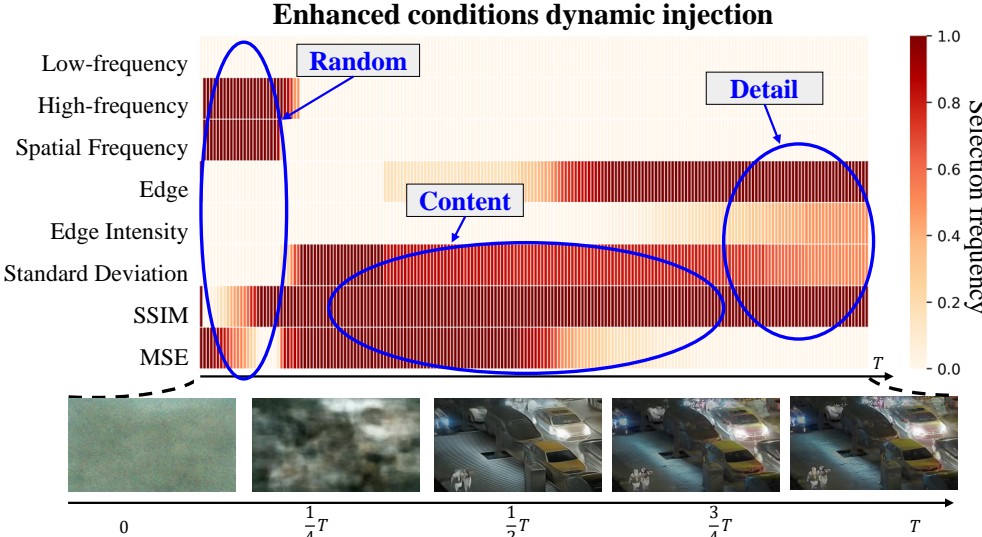

Figure 1: The conditions selection statistics during the sampling process of the LLVIP dataset. The distinct process of sampling has different favor of the conditions. The crucial role that diverse conditions play in controlling various image generation processes. Throughout the diffusion sampling, different conditions are dynamically selected to best suit the generation requirement at each stage.

generation framework, showcasing impressive capabilities in unconditional generation. Inspiringly, numerous researchers explored its controllable aspects. ILVR [22] proposed iterative latent variable refinement with a reference image to control image translation. Some recent works [23, 24] employed the diffusion model for image fusion, which fuses images in fixed fusion paradigms (fixed fusion conditions) by using its inherent reconstruction capacity. However, these approaches are not qualified for sample-customized fusion with dynamic conditions. At present, general image fusion with controllable diffusion models is still a challenging problem, warranting further exploration.

In this paper, we propose a diffusion-based controllable conditional image fusion (CCF) framework, which controls the fusion process by adaptively selecting optimization conditions. We construct a condition bank of generally used conditions, categorizing them into *basic*, *enhanced*, and *task-specific* conditions. CCF dynamically assigns fusion conditions from the condition bank and continuously injects them into the sampling process of diffusion. To enable flexible integrated conditions, we further propose a sampling-adaptive condition selection (SCS) mechanism that tailors condition selection at different denoising steps. The iterative refinements of the sampling are based on the pre-trained diffusion model without additional training. It is worth noting that the estimated fused images are conditionally controllable during the iterative denoising process. The diffusion process seamlessly integrates these conditions during the sampling process, decreasing potential impacts. As illustrated in Fig. 1, the generation process emphasizes different aspects at various sampling steps. In the initial stages, the condition selection is influenced by random noise, resulting in a random selection. During the intermediate stages, there is a shift towards content components. In the final stage, the emphasis moves to generating and selecting texture details. These various conditional factors contribute to different aspects of fusion results and demonstrate the necessity and effectiveness of introducing specific conditions in different stages. To the best of our knowledge, we for the first time propose a conditional controllable framework for image fusion. The main contributions are summarized as follows:

- We propose a pioneering conditional controllable image fusion (CCF) framework with a condition bank, achieving controllability in various image fusion scenarios and facilitating the capability of dynamic controllable image fusion.

- We propose a sampling-adaptive condition selection mechanism to subtly integrate the condition bank into denoising steps, allowing adaptive condition selection on the fly without additional training and ensuring the dynamic adaptability of the fusion process.

- Extensive experiments on various fusion tasks have confirmed our superior fusion performance against the competing methods. Furthermore, our approach qualifies for interactive manipulation of the fusion results, demonstrating our applicability and efficacy.

## 2 Related Work

Image fusion focuses on producing a unified image that amalgamates complementary information sourced from multiple source images [25].

**Specialized.** Focus on specialized tasks such as VIF, several early approaches [26–28] relied on CNNs to address challenges across various scenarios. GTF [29] defined the objective of image fusion as preserving both intensity information in infrared images and gradient information in visible images. Besides that, researchers started artificially incorporating prior knowledge to aid in the fusion process. CDDFuse [30] introduced the concept of high and low-frequency decomposition with dual-branch as prior information. Diverging from approaches tailored to single scenarios, numerous methods are now exploring the development of a unified fusion framework.DDFM [24] represents the pioneering training-free method that employs a diffusion model for multi-modal image fusion.

**Generalized.** Not limited to specialized applications, researchers aim to extend its use to generalized tasks. U2Fusion [31] introduced a unified framework capable of adaptively preserving information and facilitating joint training across various task scenarios. Additionally, SwinFusion [32] proposed a cross-domain distance learning method that has been extended to form a unified framework encompassing diverse task scenarios. Defusion [33] employs self-supervised learning techniques to decompose images and subsequently adaptively fuse them. TC-MoA [34] proposed a novel task-customized mixture of adapters for generating image fusion with a unified model, enabling adaptive prompting for various fusion tasks.

Nevertheless, these methods cannot control image fusion for adaptation to different scenarios. Therefore, we propose a method that enables image control, manipulating the fused image through existing conditions on our condition bank.

## 3 Preliminary

Denosing diffusion probabilistic models (DDPM) is a class of likelihood-based models that shows remarkable performance [35] with a stable training objective in unconditional image generation. The diffusion process entails incrementally introducing Gaussian noise to the data until it reaches a state of random noise. For a clean sample $x_0 \sim q(x_0)$ each step within the diffusion process constitutes a Markov Chain, encompassing a total of $T$ steps, relying on the data derived from the preceding step. Gaussian noise is added as follows:

$$q(x_t|x_{t-1}) = \mathcal{N}(x_t; \sqrt{1-\beta_t}x_{t-1}, \beta_t I), \tag{1}$$

where $\{\beta_t\}_{t=1}^T$ is the variance schedule of each diffusion step which is fixed and predefined. The generative process learns the inverse of the DDPM forward (diffusion) process, sampling from a distribution by reversing a gradual denoising process. We can directly sample $x_t$ at any $t$ step based on the original data $x_t \sim q(x_t|x_0)$ and via the reparameterization, it can be redefined:

$$x_t = \sqrt{\bar{\alpha}_t}x_0 + \sqrt{1-\bar{\alpha}_t}\epsilon, \tag{2}$$

where defined $\alpha_t := 1 - \beta_t$ and $\bar{\alpha} := \prod_{i=1}^t \alpha_i$. The diffusion process introduces noise to the data, whereas the inverse process represents a denoising procedure called sampling. In particular, stats with a noise $x_T \sim \mathcal{N}(0, I)$, the diffusion model learns to produce slightly less-noisy sample $x_{T-1}$, the process can be formulate by:

$$p_\theta(x_{t-1}|x_t) = \mathcal{N}(x_{t-1}; \mu_\theta(x_t, t), \Sigma_\theta(x_t, t)). \tag{3}$$

Utilizing the properties of Markov chains, decomposing $\mu_\theta$ and $\Sigma_\theta$, the process of generation is expressed as:

$$x_{t-1} = \frac{1}{\sqrt{\alpha_t}}(x_t - \frac{1-\alpha_t}{\sqrt{1-\bar{\alpha}_t}}\epsilon_\theta(x_t, t)) + \sigma_\theta^2(t)I, \tag{4}$$

where, $\sigma_\theta^2(t) = \Sigma_\theta(t) = \frac{(1-\alpha_t)(1-\bar{\alpha}_{t-1})}{1-\bar{\alpha}_t}$, $\epsilon_\theta$ signifies the output of a neural network, commonly a U-Net. This neural network predicts the noise $\epsilon_\theta$ at each step, which is utilized for the denoising procedure. It can be observed that variance is a fixed quantity, because of diffusion process parameters being constant, whereas the mean is a function dependent on $x_0$ and $x_t$. However, the stochastic process poses challenges in controlling the generative process.

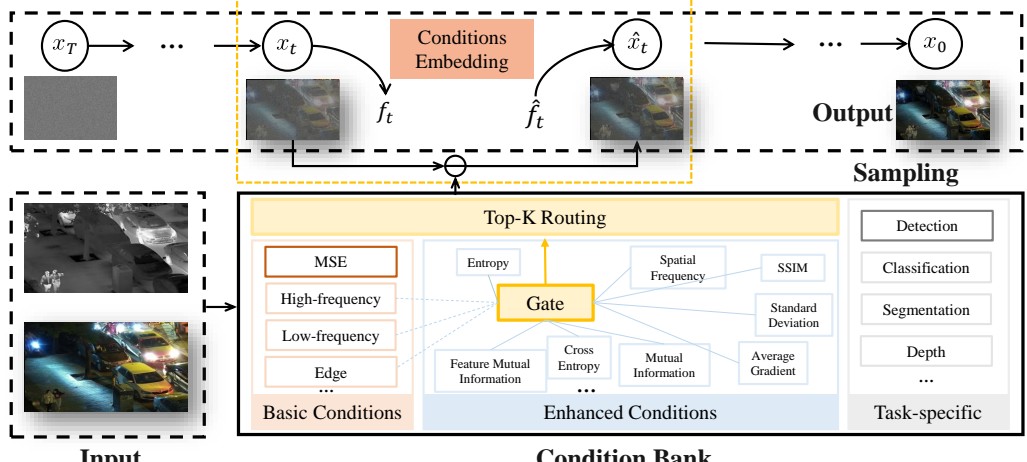

Figure 2: Illustrates the pipeline of the proposed CCF. The framework comprises two components: a sampling process utilizing a pre-trained DDPM and a condition bank with SCS.

## 4 Method

Leveraging the reconstruction capability of unconditional DDPM, we introduced a new controllable conditional image fusion (CCF) framework. Our approach accomplishes dynamically controllable image fusion via progressive condition embedding. In particular, we introduced a condition bank that regulates the incorporation of fusion information using conditions. It allows for combining the dynamic selection of multiple conditions to achieve sampling-adaptive fusion effects. As shown in Fig. 2, we illustrate our CCF framework in detail with visible-infrared image fusion (VIF). The goal is to generate a fused image $f \in \mathcal{R}^{H \times W \times N}$ from visible $v \in \mathcal{R}^{H \times W \times N}$ and infrared $i \in \mathcal{R}^{H \times W \times N}$ images, where $H$, $W$ and $N$ denote height, width, and channel numbers, respectively.

### 4.1 Controllable Conditions

Firstly, we provide the notation for the model formulation. For each sampling instance, a pre-trained DDPM represents unconditional transition $p_\theta(x_{t-1}|x_t)$. Our method facilitates the inclusion of conditional $c$ during the sampling step of unconditional transformation, without no additional training. For this purpose, we sample images from the conditional distribution $p(x_0|c)$ given condition $c$:

$$p_\theta(x_0|c) = \int dx^{(1:T)} p_\theta(x^{(0:T)}|c),$$
$$p_\theta(x^{(0:T)}|c) = p(x_T) \prod_{t=1}^{T} p_\theta(x_{t-1}|x_t, c). \tag{5}$$

Each transition $p_\theta(x_{t-1}|x_t, c)$ of the generative process depends on the condition $c$. From the property of the forward process that latent variable $x_t$ can be sampled from $x_0$ in closed-form, denoised data $x_0$ can be approximated with model prediction $\epsilon_\theta(x_t, t)$:

$$x_{0|t} \approx f_\theta(x_t, t) = \frac{(x_t - \sqrt{1 - \bar{\alpha}_t} \epsilon_\theta(x_t, t))}{\sqrt{\bar{\alpha}_t}}. \tag{6}$$

To compute $p(x_t|c)$, we can derive it from the Stochastic Differential Equation (SDE) [36]. For brevity, $x_{0|t}$ is abbreviated as $x_0$, and the expression is given by:

$$\nabla_{x_t} \log p(x_t) = -\frac{x_t - \sqrt{\bar{\alpha}_t} x_0}{1 - \bar{\alpha}_t}. \tag{7}$$

Classifier Guidance [37] can be intuitively elucidated via the score function, which logarithmically decomposes the conditional generation probability using Bayes' theorem:

$$\nabla \log p(x_t|c) = \nabla \log p(x_t) + \nabla \log p(c|x_t). \tag{8}$$

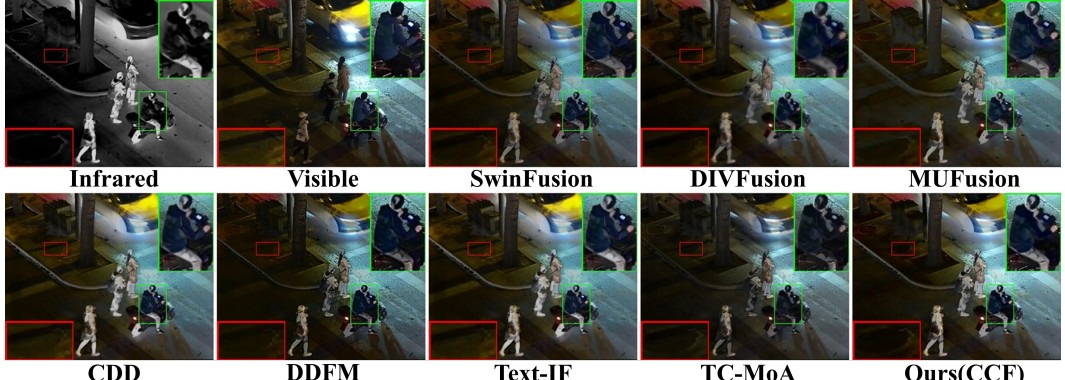

| Infrared | Visible | SwinFusion | DIVFusion | MUFusion |
| CDD | DDFM | Text-IF | TC-MoA | Ours(CCF) |

Figure 3: Qualitative comparisons of our CCF and the competing methods on VIF fusion task of LLVIP dataset.

Diffusion posterior sampling can yield a more favorable generative trajectory, especially in noisy settings, [38] estimation $\log p(c|x_t)$ as follows:

$$\nabla \log p(c|x_t) \approx \nabla \log p(c|\hat{x}_0). \tag{9}$$

Further, as demonstrated in [39], we can express the score function as:

$$\nabla \log p(c|x_t) \approx \nabla \log p(c|\hat{x}_0) = f_\theta(x_t, t) - \lambda \nabla ||C - \mathcal{A}(x_0)||_2. \tag{10}$$

Here, $\mathcal{A}(\cdot)$ can be linear or nonlinear operation. We represent $||C - \mathcal{A}(x_0)||$ as $\delta_C$. Now, the objective is to obtain $\hat{x}_0$ and incorporate the condition into it. We can minimize $\delta_C$ to regulate the sampling within the diffusion process. In the following section, we will provide a detailed explanation of how to build a condition bank and how to select conditions.

## 4.2 Condition Bank

We empirically construct a *condition bank* and divide the image constraints into three categories: basic fusion conditions, enhanced fusion conditions, and task-specific fusion conditions. Basic fusion conditions are utilized throughout the entire sampling process, while enhanced fusion conditions are dynamically selected. Task-specific fusion conditions are manually optional, tailored to specific tasks, and may possess unique attributes that can be customized for various task scenarios. All conditions can be part of the enhanced condition set, enabling dynamic selection. The condition bank presented in this paper includes some common conditions, but additional conditions can be explored and utilized in other scenarios.

In the above formulation, each conditional Markov transition with the given condition $c$ is shown in Eq. 5. In particular, we constructed a condition bank that allows us to select required conditions $C = \{c_1, c_2, ..., c_n\}$, subsequently integrating them into the unconditional DDPM for executing conditional image fusion. Let $C$ represent a condition bank comprising a series of conditions. The function $\delta_C$ represents the difference between the source images with the given condition. In every sampling step $t$, the difference function $\delta_{c_i}$ can be minimized using gradient descent. These conditions help regulate the image information within each modality involved in the fusion process.

**Basic Conditions**. As shown in Fig. 2, basic conditions are essential to select for a basic generation. The basic conditions aim to synthesize a foundational fused image, offering an initial, coarse representation. The fused image serves as a primary fusion output, capturing essential features from the source images, though it may suffer from detail loss or texture blur. Notably, different scenarios may require adjustments to the basic condition, as the specific requirements of each fusion task, such as clarity, contrast, and other priorities, can influence its selection. Tailoring the basic condition to align with the unique demands of each task thus ensures an effective fusion process.

**Enhanced Conditions**. Besides basic conditions, we added enhancement conditions for refining the image generation process. The condition bank contains a variety of enhanced conditions, inspired by various Image Quality Assessments (IQA) such as SSIM, and standard deviation(SD). These conditions can be integrated into the CCF generation process to improve the quality of the generated images. The enhanced conditions can be selected with SCS algorithm, allowing different steps of the diffusion sampling process to be optimized with different conditions. This targeted approach ensures

that each phase of the image generation is adapted to the specific requirements of that stage, resulting in higher quality fused images.

**Task-specific Conditions**. The purpose of image fusion is to facilitate various downstream tasks. To meet the specific requirements of these tasks, we offer task-specific conditions. These conditions can be manually added to ensure that the fusion process is tailored to suit the downstream applications better. For example, a detection condition can be introduced by utilizing the feature extracted by a detection network $F = D(x)$. The detection condition is formulated as $||F(x), F(M)||_2$, where $X \in \{x_0\}_i^m$ and $M$ is the set of $m$ modalities. Other task-specific conditions can be similarly tailored to optimize the fusion process for different tasks. By integrating these task-specific conditions, the image fusion process can be precisely aligned with the demands of various applications, enhancing the effectiveness and utility of the generated images.

### 4.3 Sampling-adaptive Condition Selection

During the diffusion sampling process, it is crucial to focus on generating distinct aspects of the image. To address this, we designed an algorithm that dynamically selects the appropriate condition from the condition bank to fit each sampling stage. This selection process can be denoted as $C_{opt} = \text{Top}_k\{\text{Gate}(C)\}$. The $C_{opt}$ is the selected condition, and Gate is the gate of selection. We hypothesize that rapidly changing conditions during the sampling process should be prioritized as they indicate greater significance at that generation stage. Inspired by multi-task learning [40], the gate of conditions can be calculated using the following formula:

$$\text{Gate} = [\omega_1, ..\omega_c], \omega_i(t) = \omega_i(t-1) - \bigtriangledown\omega_i. \tag{11}$$

The $\omega_i(t-1)$ represents the condition gradient from the previous step, and the $\bigtriangledown\omega_i$ is calculated as:

$$\bigtriangledown\omega_i = \sum_i |G_W^i(t) - \mathbb{E}[G_W^i(t)] \times \alpha_i(t)|_1, \tag{12}$$

where, $G_W = ||\omega_i(t)L_i(t)||_2$, $L_i(t)$ represents the condition gradient at step $t$, and the value can be calculated using a gradient descent algorithm in minimize $\delta_{c_i}$. The $\alpha_i(t)$ is defined as:

$$\alpha_i(t) = [\frac{\widetilde{L}_i}{\mathbb{E}[\widetilde{L}_i]}]^\theta, \tag{13}$$

where, $\widetilde{L}_i = \frac{L_i(t)}{L_i(t-1)}$, $\theta$ is a hyper-parameter. By incorporating this SCS algorithm, we can efficiently choose the most relevant condition for each step in the diffusion process, thereby enhancing the quality of the conditional image fusion.

## 5 Experiments

**Datasets.** We conduct experiments in three image fusion scenarios: multi-modal, multi-focus, and multi-exposure image fusion. For multi-modal image fusion task, we conducted experiments on the LLVIP [41] dataset and referred to the test set outlined in Zhu et al. [34]. For MEF and MFF, our testing procedure followed the test setting in MFFW dataset [42] and MEFB dataset [43], respectively. Additionally, we test our method on the TNO dataset and Harvard medical dataset to assess our method's performance within the multi-modal fusion domain, detailed in App. B and H.

**Implementation Details.** Our method utilizes a pre-trained diffusion model as our foundational model [44]. This model was directly applied without any subsequent fine-tuning for specific task requirements during our experiments. The experiments are conducted on Huawei Atlas 800 Training Server with CANN and NVIDIA RTX 3090 GPU. Experimental settings are shown in App. A.

**Evaluation Metrics.** We evaluated the fusion results in both quantitative and qualitative. Qualitative evaluation primarily hinges on subjective visual assessments conducted by individuals. We expect that the fused image will exhibit rich texture details and abundant color saturability. Objective evaluation primarily focuses on measuring the quality assessments of individual fused images and their deviations from the source images. For different task scenarios, the different evaluation metrics used, specifically, we employ six metrics including Structural Similarity (SSIM), Mean squared error (MSE), correlation coefficient (CC), peak signal-to-noise ratio (PSNR), modified fusion artifacts measure (Nabf). In the MFF and MEF tasks, considering the different task scenarios, we employ standard deviation (SD), average gradient (AG), spatial frequency (SF), and sum of the correlations of differences (SCD) for evaluation metrics.

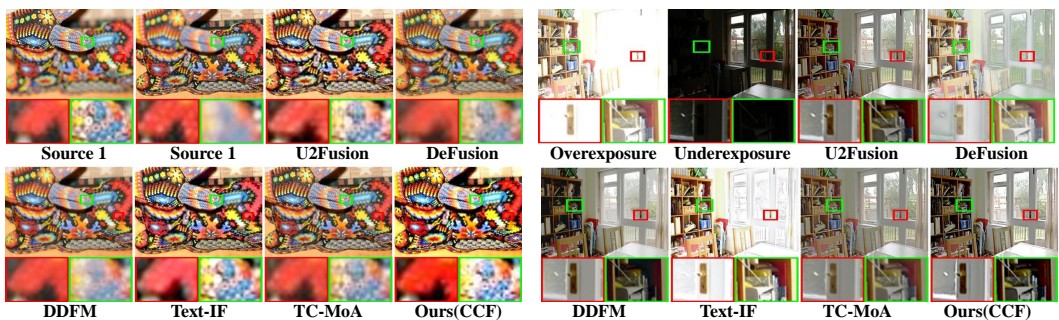

Figure 4: Qualitative comparisons of various methods in MFF task from MFFW dataset.

## 5.1 Evaluation on Multi-Modal Image Fusion

For multi-modal image fusion, we compare our method with the state-of-the-art methods: Swin-Fusion [32], DIVFusion [45], MUFusion [46], CDDFuse [30], DDFM [24], Text-IF [47], and TC-MoA [34] and on the LLVIP dataset. More datasets and comparison methods are shown in App. B. Note that our method, like DDFM, does not require additional tuning.

**Quantitative Comparisons.** We employ five quantitative metrics to evaluate our model, as shown in Table 1. Our method demonstrated exceptional performance across various evaluation metrics. On the LLVIP dataset, our method achieved the best results in SSIM, MSE, and CC indicators. Specifically, our method outperformed others in SSIM and CC, with improvements of 0.02 and 0.035 over the second-best results, respectively. Additionally, lower MSE values indi-

Table 1: Comparison with SOTAs in the LLVIP dataset. The red/blue/green indicates the best, runner-up and third best.

| | SSIM↑ | MSE↓ | CC↑ | PSNR↑ | Nabf↓ |
|---|---|---|---|---|---|
| | **LLVIP Dataset** | | | | |
| SwinFusion [32] | 0.81 | 2845 | 0.668 | 32.33 | 0.023 |
| DIVFusion [45] | 0.82 | 6450 | 0.655 | 21.60 | 0.044 |
| MUFusion [46] | 1.10 | 2069 | 0.648 | 31.64 | 0.030 |
| CDDFuse [30] | 1.18 | 2545 | 0.670 | 32.13 | 0.016 |
| DDFM [24] | 1.18 | 2056 | 0.668 | 36.10 | 0.004 |
| Text-IF [47] | 1.20 | 2135 | 0.669 | 31.97 | 0.023 |
| TC-MoA [34] | 1.20 | 2790 | 0.666 | 33.00 | 0.017 |
| CCF (ours) | 1.22 | 1694 | 0.705 | 32.71 | 0.005 |

cate better performance, with our method showing a reduction of 362 compared to the second-best methods in these metrics. This indicates that our method retains more information from the source images. The results show suboptimal performance in Nabf but are close to the best values, indicating the fused image with less noise. Notably, our method does not necessitate turning and holds its ground against methods requiring training. Compared to existing LLM tuning methods, our model performs slightly worse in terms of PSNR. This demonstrates the excellent performance of our model, achieving high performance across almost all indicators in a tuning-free model.

**Qualitative Comparisons.** Furthermore, the incorporation of the basic condition and enhanced conditions enables effective preservation of the background and texture. This comparison underscores our model's efficacy in image fusion, resulting in outstanding visual outcomes. As shown in Fig. 3, our method showcases superior visual quality compared to other approaches. Specifically, our method excels in preserving intricate texture details well lid in low light (Fig. 3 red box). Although TC-MoA and MUFusion approach our method in retaining details, they exhibit visible artifacts, blur, and low contrast—characteristics absent in CCF (Fig. 3, green box). CCF exhibits the highest contrast, the clearest details, and the most information content, further highlighting its superiority in preserving texture details. Its excellent detail retention and clear background generation further demonstrate the effectiveness of our proposed method.

## 5.2 Evaluation on Multi-Focus Fusion

For multi-focus image fusion, we compare our CCF with five general image fusion methods: U2Fusion [31], DeFusion [33], DDFM [24], Text-IF [47], and TC-MoA [34].

**Quantitative Comparisons.** We employ four quantitative metrics to evaluate our model, as shown in Table 2 (left). Our method significantly outperforms the comparison methods, achieving SOTA across all metrics. Specifically, the SD is 11.19 higher than the suboptimal value, indicating higher

Table 2: Comparison with SOTAs. The **red**/**blue**/**green** indicates the best, runner-up and third best.

| | **MFFW Dataset** | | | | **MEFB Dataset** | | | |
| --- | --- | --- | --- | --- | --- | --- | --- | --- |
| | SD ↑ | AG ↑ | SF ↑ | SCD ↑ | SD ↑ | AG ↑ | SF ↑ | SCD ↑ |
| U2Fusion [31] | **67.83** | **8.08** | **22.19** | **2.67** | **64.88** | **5.56** | **18.74** | **2.05** |
| DeFusion [33] | 54.75 | 4.76 | 12.72 | 0.50 | 52.75 | 4.32 | 14.12 | −0.97 |
| DDFM [24] | 56.34 | 4.47 | 12.21 | 0.90 | **67.30** | 3.82 | 13.40 | **2.12** |
| Text-IF [47] | **66.27** | **7.72** | **21.58** | **2.27** | 62.51 | 4.78 | **17.26** | 1.46 |
| TC-MoA [34] | 57.55 | 6.95 | 20.67 | 1.03 | 50.27 | **4.82** | 15.64 | 0.42 |
| CCF (ours) | **79.02** | **8.18** | **24.30** | **2.78** | **71.88** | **5.70** | **20.28** | **3.00** |

contrast and clearer images. The SCD is 0.11 higher than the suboptimal value, suggesting a lower error between source images and fused images. The AG and SF also rank first demonstrating the retention of more texture details. These results showcase that our method effectively preserves details from the source images and produces high-quality fused images.

**Qualitative Comparisons.** As illustrated in Fig. 4, our proposed method demonstrates outstanding visual performance, particularly in preserving intricate details. We have carefully selected specific conditions that allow our approach to effectively handle the blurring caused by multi-focus scenes while retaining the original image's lighting and color information. In comparison, other DDPM-based methods such as DDFM are unable to achieve the same level of effectiveness as our approach. In Fig. 4 (red), our method excels in preserving the details of the watch hand. The closest result to ours is U2Fusion, but it loses texture and color fidelity, appearing blurry, in Fig. 4 (green). In short, our method performs well in maintaining both color and authentic details.

## 5.3 Evaluation on Multi-Exposure Fusion

For multi-exposure image fusion, we compare our model with five general image fusion methods, i.e., U2Fusion [31], DeFusion [33], DDFM [24], Text-IF [47], and TC-MoA [34].

**Quantitative Comparisons.** As demonstrated in Table 2 (right), our method achieved SOTA on the MEF index, analogous to the results observed for the MFF task. Notably, the metrics SD, AG, and SF signify the highest image quality, while SCD exhibits the highest correlation. Each of these metrics attained state-of-the-art performance levels, underscoring the efficacy of our approach. This consistent performance across multiple metrics illustrates the robustness and versatility of our method in enhancing image quality and fidelity.

**Qualitative Comparisons.** Fig. 4 demonstrates the excellent visual performance of our method. Our approach effectively addresses the issue of overexposure while preserving crucial details. In contrast, DDFM, which relies on finding the middle value of two images, struggles to maintain texture details. Similarly, Text-IF tends to result in higher average brightness, which can lead to content loss in overexposed scenes. Defusion and TC-MoA exhibit similar visual performance, with more blurred edges compared to our method. In comparison, our method strikes a balance between these challenges, resulting in superior visual fidelity and saturation compared to other existing methods.

## 5.4 Task-specific Conditions

The task-specific conditions can be manually selected. In this section, we use the detection condition as an example. The detection model employed is YOLOv5 [48], trained on the LLVIP dataset. We randomly select 287 images from the LLVIP test dataset to validate our method with the detection condition. Before adding the detection condition, the fused image achieved mAP.5 = 0.737, mAP.5 : .95 = 0.509, and a recall of 0.737. After incorporating the detection condition, the mAP.5 increased by 0.049 to 0.907, mAP.5 : .95 increased by 0.054 to 0.563, and recall significantly improved to 0.832. Fig. 5 visualizes several cases detected using YOLOv5.

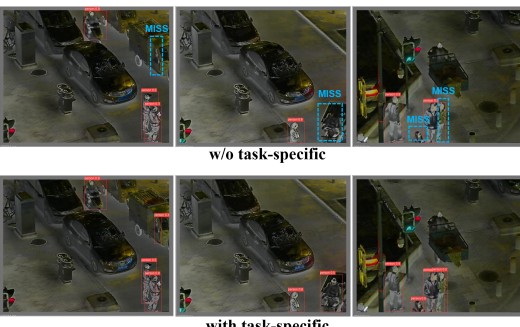

w/o task-specific

with task-specific

Figure 5: The visualization of the w/o and with task-specific conditions.

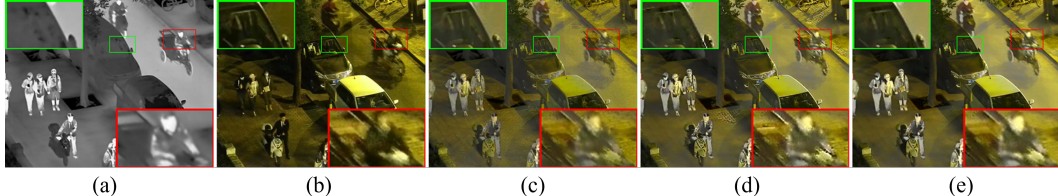

(a)          (b)          (c)          (d)          (e)

Figure 6: Ablation study of condition bank. (a) is infrared image, (b) is visible image, (c) is basic condition, (d) is CCF w/o SCS, (e) is CCF.

The fused images with the detection condition exhibit higher confidence (columns 1 and 2) and recall (columns 1, 2, and 3). This demonstrates that task-specific conditions can enhance the performance of fused images in downstream tasks. By integrating task-specific conditions, the fused image can be precisely aligned with the demands of various applications, enhancing the overall effectiveness and utility of the generated images. More detail shows in App. F.

### 5.5 Ablation Study

Numerous ablation experiments were conducted to assess the effectiveness of different components of our model. The aforementioned four metrics were utilized to evaluate fusion performance across different experimental configurations. The quantitative results are presented in App. G.

To validate the effectiveness of the condition bank, we systematically add basic and enhanced conditions individually and then verify the effectiveness of SCS. Fig. 6 illustrates a gradual variation in performance metrics with the progressive addition of conditions.

In Fig. 6(a), using only the basic condition results in image fusion. When all enhanced conditions are injected together without SCS, the metrics all reduced (Fig. 6(b)), likely due to the conflict between different conditions, as evidenced by the messy lines and random noise in Fig. 6(d). This demonstrates that injecting conditions unevenly without DCS leads to suboptimal results. After introducing DCS, both the metrics and visual results (Fig. 6(e)) are well-balanced, with conditions being injected as needed during the generation process. This indicates the effectiveness of the condition bank and DCS in dynamically and appropriately selecting the conditions.

### 5.6 Discussion on Controllable Fusion

Our method enables the dynamic selection of conditions, allowing for the adaptive customization of conditions for each sample. As illustrated in Fig. 1, the statistics show the condition selection at various diffusion sampling stages. Initially, the process is stochastic due to random noise (around 0 to $\frac{1}{4}T$ steps). As denoising progresses, the content of the image starts to be dominant in fusion, and conditions like MSE, SSIM, and SD are most frequently chosen (around $\frac{1}{4}T$ to $\frac{3}{4}T$ steps). As the process continues, and the fusion model tends to generate more texture details, conditions like EI and edge become more dominant (around $\frac{3}{4}T$ to $T$ steps). SSIM also remains selected to constrain the structure, whereas the frequency of SD selection decreases. This demonstrates the crucial role of dynamically selecting conditions during the diffusion sampling process. Our CCF adaptively decomposes diverse conflict fusion conditions into different denoising steps, which is significantly compatible with the reconstruction preferences of different steps in the denoising model. This dynamic fusion paradigm ensures appropriate condition selection at each stage of the sampling.

## 6 Conclusion

In this paper, we introduce a learning-free approach for conditional controllable image fusion (CCF), utilizing a condition bank to regulate joint information with a pre-trained DDPM. We capitalize on the remarkable reconstruction abilities of DDPM and integrate them into the sampling steps. Sample-adaptive condition selection facilitates fusion in dynamic scenarios. Varied fusion images can personalize their conditions to emphasize different aspects. Empirical findings demonstrate that CCF surpasses the competing methods in achieving superior performance for general image fusion tasks. In the future, we will further explore automatic manners to distinguish basic and enhanced conditions to reduce empirical intervention, thereby enabling more robust and reliable image fusion.

**Acknowledgements.**

This work was sponsored by National Science and Technology Major Project (No. 2022ZD0116500), National Natural Science Foundation of China (No.s 62476198, 62436002, 62222608, U23B2049, 62106171 and 61925602), Tianjin Natural Science Funds for Distinguished Young Scholar (No. 23JCJQJC00270), the Zhejiang Provincial Natural Science Foundation of China (No. LD24F020004), and CCF-Baidu Open Fund. This work was also sponsored by CAAI-CANN Open Fund, developed on OpenI Community.

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

# Appendix

## A   Experimental Settings

Our method not only excels in individual settings to generate customized images but also demonstrates robust performance in general settings. In this section, we elaborate on numerous experiments for image fusion tasks to demonstrate the superiority of our method. For different tasks, the basic conditions vary. Specifically, in the VIF task, MSE [24, 46, 31, 34] serves as the basic condition, where $c_{\mathrm{MSE}}$ can be expressed as $||\mathrm{MSE}(i, x_0) + \mathrm{MSE}(v, x_0)||$. In contrast, for the MEF and MFF tasks, to achieve clearer and higher fidelity images and inspired by [29, 6, 49, 50], the basic conditions include MSE, frequency and edge conditions, as detailed in the App. C. To ensure convenience and fairness, we select eight enhanced conditions from the condition bank: SSIM, MSE, Edge, Low-frequency, High-frequency, Spatial Frequency, Edge Intensity, and Standard Deviation while setting $k = 3$ across all tasks.

## B   Experiments on More Comparisons

In our evaluation using the TNO dataset, we referred to the test set outlined in the work by Tang et al. [51] We evaluated the fusion results in both quantitative and qualitative. Qualitative evaluation primarily hinges on subjective visual assessments conducted by individuals. We expect that the fused image will exhibit rich texture details and abundant color saturability. For different task scenarios, the different evaluation metrics used, specifically, we employ six metrics including standard deviation (SD), entropy (EN), spatial frequency (SF), sum of the correlations of differences (SCD), Structural Similarity (SSIM), and edge intensity (EI). In the MFF and MEF tasks, considering the different task scenarios, we employ SD, AG, SF, SCD, and MSE for evaluation metrics. For multi-modal image fusion, we compare our method with four task-specific methods: DenseFuse [52], RFN-Nest [53], UMF-CMGR [54], YDTR [55], and three general approaches U2Fusion [31], DeFusion [33]and DDFM [24] on the LLVIP dataset and TNO dataset.

**Quantitative Comparisons.** We employ 6 quantitative metrics to evaluate our model, as shown in Table 3. Our method demonstrated exceptional performance across various evaluation metrics. On the TNO dataset, our method achieves the best result on the SD, EN, and SCD indicators. The result shows suboptimal performance in SF and is close to the best values. Notably, our method does not necessitate turning and holds its ground against methods requiring training. Compared with the learning-free DDFM and DeFusion, our method shows better results. This demonstrates the excellent performance of our model, achieving high performance across almost all indicators in a tuning-free model.

**Qualitative Comparisons.** The incorporation of both basic and enhanced conditions enables effective preservation of background and texture. This comparison underscores our model's efficacy in image fusion, resulting in outstanding visual outcomes. As shown in Fig.7, our method demonstrates superior visual quality compared to other approaches. Specifically, it excels in preserving intricate texture details and color fidelity, such as license plate numbers (highlighted in the red box in Fig.7). The DDFM and DeFusion approaches retain fewer texture details. As depicted in Fig. 7, CCF exhibits the highest contrast, the clearest details, and the most comprehensive information content, further highlighting its superiority in preserving texture details. Its excellent detail retention ability and clear background generation further prove the effectiveness of our proposed method.

## C   Basic Conditions of the MEF and MFF

**High-frequency** It's commonly understood that the high-frequency information in both modalities is distinctive to each modality. For instance, texture and detailed information are specific to visible images, while thermal radiation information pertains to infrared images. Specifically, we utilizes wavelet transform to extract high-frequency information from the image. For example, 2D discrete wavelet transformation with Haar wavelets to transform the input image (IM) into four sub-bands can be expressed as:

$$\{LL_k, HL_k, LH_k, HH_k\} = \mathcal{W}_k(LL_{k-1}) \tag{14}$$

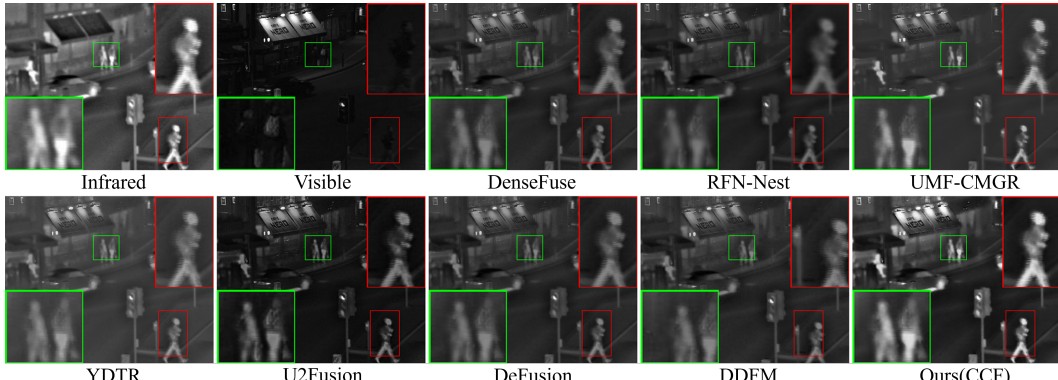

Figure 7: Qualitative comparisons of our CCF and the competing methods on VIF fusion task of TNO dataset

where $LL_k, HL_k, LH_k, HH_k \in \mathcal{R}^{\frac{H}{2^k} \times \frac{W}{2^k} \times c}, k \in [1, K]$, signify the averages of the input image and the high-frequency details in the vertical, horizontal, and diagonal directions, respectively, at the first-level wavelet decomposition. Denote that the high-frequency details $H_F(\text{IM})$ from IM. So we can see the high-frequency condition below:

$$\widetilde{HF} = H_F(x_0) - F(\lambda_h|H_F(i)|, (1 - \lambda_h)|H_F(v)|) \tag{15}$$

where $|\cdot|$ stands for the absolute operation and $F(\cdot, \cdot)$ can represent a variety of customized functions. In this paper, specifically, we employ the max function. The restored average coefficient and the high-frequency coefficient at scale $k$ are correspondingly converted into the output at scale $k - 1$ by employing $\mathcal{W}^{-1}$ the 2D inverse discrete wavelet transform:

$$\delta_{c_h} = \mathcal{W}^{-1}\{LL_k, \widetilde{HF}_k\} \tag{16}$$

**Low-Frequency Condition**. Low-frequency components generally encapsulate information common to both modes. For simplicity, we directly utilize the low-frequency information obtained via wavelet transform. However, each mode may possess slightly varying amounts of low-frequency information. Our conditions allow for the adjustment of the low-frequency $L_F$ fusion ratio with hyperparameters, i.e.,

$$\widetilde{LL} = L_F(x_0) - (\lambda_l L_F(i) + (1 - \lambda_l)L_F(v)) \tag{17}$$

$$\delta_{c_l} = \mathcal{W}^{-1}\{\widetilde{LL}_k, HF_k\} \tag{18}$$

**Edge Condition**. We aim at the fused image to retain the more intricate texture details from both infrared and visible images. To achieve this, we integrated edge conditions into the set of conditions

Table 3: Comparison with SOTAs. **Red** indicates the best, **blue** indicates the second best, and **green** indicates the third best.

| | **TNO Dataset** | | | | | |
|---|---|---|---|---|---|---|
| | SD ↑ | EN ↑ | SF ↑ | SSIM ↑ | SCD ↑ | EI↑ |
| DenseFuse [52] | 34.83 | 6.82 | **8.99** | 1.38 | 1.78 | **8.75** |
| RFN-Nest [53] | **36.90** | **6.96** | 5.87 | 1.31 | **1.78** | 7.13 |
| UMF-CMGR [54] | 29.97 | 6.53 | 8.17 | **1.43** | 1.64 | 7.32 |
| YDTR [55] | 28.03 | 6.43 | 7.62 | **1.41** | 1.56 | 6.81 |
| U2Fusion [56] | **37.70** | **7.00** | **11.86** | 1.28 | **1.78** | **12.78** |
| DeFusion [31] | 30.38 | 6.58 | 6.20 | 1.38 | 1.53 | 6.54 |
| DDFM [24] | 34.45 | 6.85 | 7.27 | 1.08 | 1.54 | 7.89 |
| CCF (ours) | **40.11** | **7.01** | **10.20** | **1.38** | **1.84** | **8.29** |

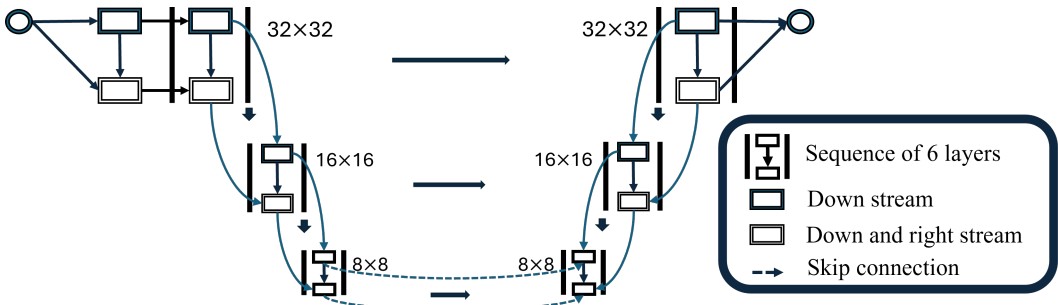

Figure 8: Framework of our neural network architecture.

to enhance the edge texture in the fused image. The condition is defined as :

$$\delta_{c_e} = |\nabla(x_0)| - F(\lambda_e|\nabla(i)|, (1-\lambda_e)|\nabla(v)|) \tag{19}$$

where $\nabla$ indicates the Sobel gradient operator, which measures the texture detail information of an image.

**MSE Condition**. We incorporate the information from the original image into the fused image as a supplementary content condition. Essentially, we utilize the image reconstruction capability of DDPM to supplement lost details and enhance the informational content pertaining to the target in the fused image, i.e.

$$\delta_{c_{MSE}} = x_0 - \Psi^{-1}(\lambda_{MSE}\Psi(x_0) - (1-\lambda_{MSE})\Psi(y)) \tag{20}$$

where $\Psi$ and $\Psi^{-1}$ represent the downsample and upsample operator, respectively, while $y$ can refer to either the infrared or visible image.

Totally, these conditions can be combined to govern the generation of the final fused image:

$$\delta_c = \eta_h\delta_{c_h} + \eta_l\delta_{c_l} + \eta_e\delta_{c_e} + \eta_{cont}\delta_{c_{MSE}} \tag{21}$$

## D More Details of CCF

In the sampling process, the neural network architecture is similar to PixelCNN++ [57], consisting of a U-Net backbone with group normalization. The diffusion step $t$ is defined by incorporating the Transformer [58] sinusoidal position embedding into each residual block. As shown in Fig. 8, six feature map resolutions are utilized, with two convolutional residual blocks per resolution level. Additionally, self-attention blocks are incorporated at the $16{\times}16$ resolution between the convolutional blocks.

Representing a process of reverse diffusion (sampling), starting from a standard normal distribution $\mathcal{N}(0, I)$, $T$ sampling steps are performed

---

**Algorithm 1** CCF

**Input:** : $i,v$
**Output:** : $f$
1: $\delta_C \leftarrow C(i, v)$
2: Sample $x_T \sim \mathcal{N}(0, I)$
3: **for** t=T, .., 1 **do**
4:    $z \sim \mathcal{N}(0, I)$
5:    $x_0 = \frac{x_t - \sqrt{1-\bar{\alpha}_t}\epsilon_\theta(x_{t-1}|x_t)}{\sqrt{\bar{\alpha}_t}}$
6:    **for** k=N,..., 1 **do**
7:       $\hat{x}_0 \leftarrow x_0 - \lambda \bigtriangledown \delta_{c_k}$
8:    **end for**
9: **end for**

---

to generate the final fused image. However, the generation with unconditional DDPM [35] is uncontrollable. Therefore, conditions selected from the condition bank we constructed are introduced to control the sampling process. More precisely, the conditions can rectify each step's estimation of $x_0$. Finally, after the $T$ step, the final fused image can be generated. The algorithm of CCF is presented in Algorithm 1.

## E More About Enhanced Conditions

We follow the recent works [30, 24, 46], and adopt the eight most commonly used constraints as conditions in this paper. However, as shown in Table 4, our approach is not restricted to these

Table 4: Comparison across different numbers of enhancement conditions. For the 4 conditions, conditions include SSIM, MSE, Edge, and SD. The 8 conditions expand to include SSIM, MSE, Edge, SD, Low-frequency, High-frequency, SF, and EI. The 12 conditions incorporate the previous 8 conditions with four additional conditions: CC, MMS-SSIM, SCD, and VIF. **Bold** indicates the best results

| Number of conditions | SSIM ↑ | MSE ↓ | CC ↑ | PSNR ↑ | Nabf ↓ | Avg. Runtime(s) |
|---|---|---|---|---|---|---|
| 4 | 1.12 | 1820 | 0.699 | 32.20 | 0.034 | 188 |
| 8 (ours) | **1.22** | 1694 | 0.705 | 32.58 | 0.005 | 222 |
| 12 | 1.20 | **1545** | **0.719** | **33.21** | **0.002** | 454 |

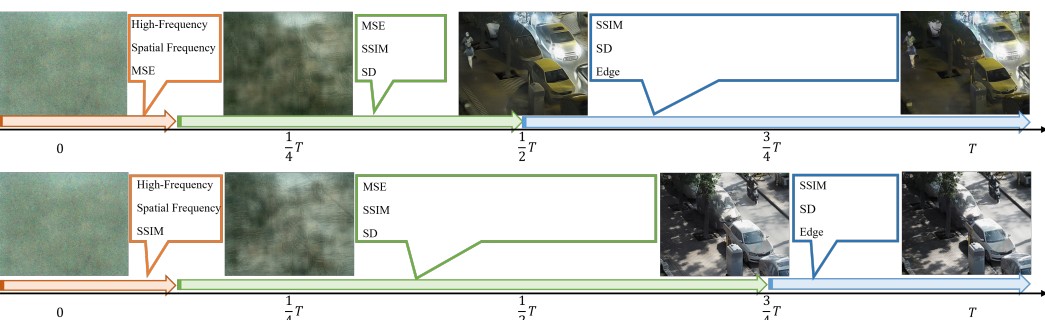

Figure 9: Example of enhanced conditions selection in denoising iteration for rapidly changing scenarios.

constraints; fewer conditions can be randomly selected (here we select SSIM, MSE, Edge, and SD), and additional enhanced conditions [31, 34](Correlation Coefficient (CC), Multi-Scale Structural Similarity (MS-SSIM), the Sum of the Correlations of Differences (SCD), and the Visual Information Fidelity (VIF)) can also be incorporated. Additionally, we evaluate the runtime across different numbers of conditions to assess their impact on efficiency. While adding more conditions slightly improves performance, it also increases inference runtime.

Fig. 9 illustrates the selection of enhanced conditions at each denoising step in the same location under both daylight and nighttime scenarios, representing a typical rapidly changing environment. It demonstrates that CCF can adapt to these changing environments by selecting different conditions at various denoising steps to respond effectively to dynamic environmental characteristics.

## F More About Task-specific Conditions

The task-specific conditions are incorporated to guide the fusion process throughout the denoising procedure. For instance, we take the Euclidean distance of features extracted by the object detection model as the detection condition. In our experiments, we employ YOLOv5, which is pretrained on the visible modality, to extract features from both the estimated $x_0$ at each step and the visible image. The Euclidean distance is minimized in the inverse diffusion process iteratively. Consequently, the final fused image progressively integrates the object-specific information, enhancing the fusion performance. We provide additional visualizations of the detection conditions, as shown in Fig. 10, directly fused images appear to have a lot of missed detection in raw 1-4 and the detection scores are generally higher. The last row of Fig.10 demonstrates a reduction in false-positive boxes when the detection condition is applied.

Table 5 also supports this, indicating that metrics such as mAP.5, mAP.5:95, and recall have increased remarkably, showing that our method can customize conditions to fit the downstream task effectively. By tailoring conditions to specific tasks, our approach effectively improves the applicability of fused images for particular applications, as evidenced by the improved detection metrics. We further explored the use of different levels of features obtained with YoloV5 as detection conditions. Specifically, we evaluate in three scales and each of the scales is evaluated individually, and all features combined involve using all three scale features simultaneously, averaged, to add task-specific perception to enhance task performance. Furthermore, Table 5 indicates that there is no significant

Table 5: Comparison of task-specific conditions across different scales of features in YOLOv5s.The **Bold** indicates the best results.

| Scale of feature | Position | mAP.5 | mAP.5:95 | Recall |
|---|---|---|---|---|
| w/o Feature | - | 0.858 | 0.509 | 0.737 |
| Feature 1 (160×128) | Before | 0.884 | 0.537 | 0.832 |
| Feature 2 (80×64) | Before | 0.890 | 0.540 | 0.826 |
| Feature 3 (40×32) (ours) | Before | **0.907** | **0.563** | 0.832 |
| All Features ($\frac{1}{3}\sum_3^i$ Feature $i$) | Before | 0.882 | 0.541 | **0.837** |
| Final performance | After | 0.894 | 0.537 | 0.824 |

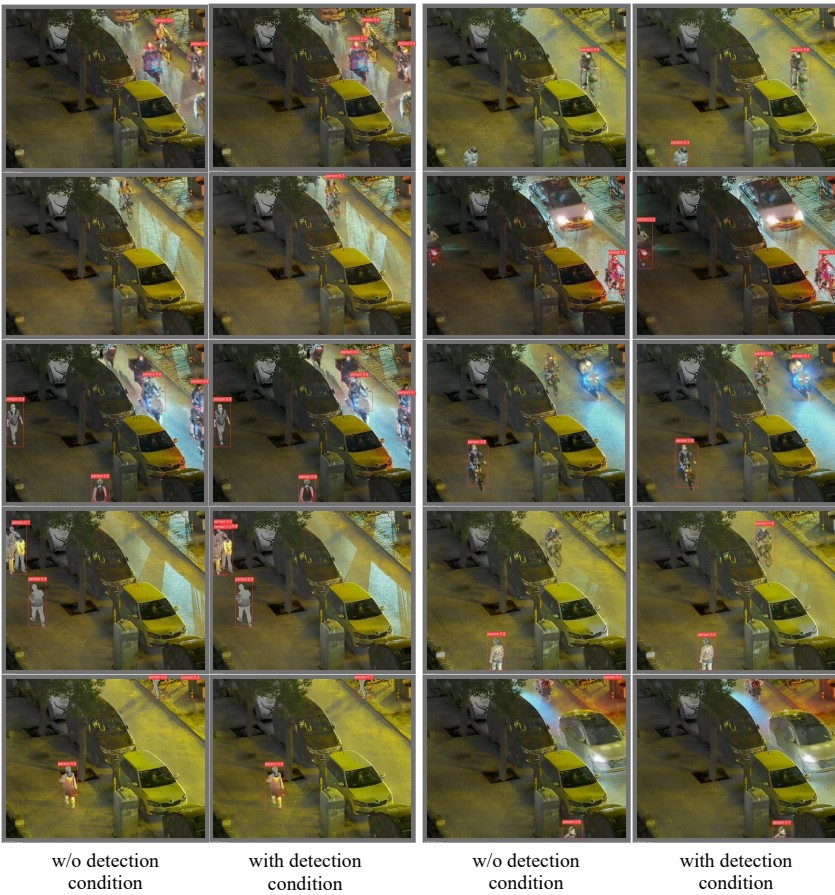

| w/o detection condition | with detection condition | w/o detection condition | with detection condition |
|---|---|---|---|

Figure 10: Qualitative Comparisons of w/o and with the detection condition.

difference between using different features. The constraint primarily enhances the implicit expressive ability of the image, enabling it to better adapt to downstream tasks.

# G    Ablation

The quantitative results of the ablation study compare different scenarios: without DDPM, with only basic conditions, with enhanced conditions but without SCS, and the complete CCF results. With the reconstruction capability of DDPM, the fusion performance significantly improved after the conditions were integrated. Some metrics for the enhanced conditions without SCS show a decline due to conflicts among the various focusing aspect conditions. However, after incorporating SCS, most metrics improve, demonstrating that SCS can enhance the image fusion process, producing high-quality images.

Table 6: Ablation studies on the LLVIP dataset. **Bold** indicates the best.

| DDPM | Basic Conditions | Enhanced conditions | SCS | SSIM↑ | MSE↓ | CC↑ | PSNR↑ | Nabf↓ |
|:---:|:---:|:---:|:---:|:---:|:---:|:---:|:---:|:---:|
| ✓ | | | | 0.28 | 2947 | 0.452 | 27.64 | 0.131 |
| ✓ | ✓ | | | 1.16 | 1785 | 0.683 | **33.06** | 0.032 |
| ✓ | ✓ | ✓ | | 1.16 | 1779 | 0.677 | 32.17 | 0.072 |
| ✓ | ✓ | ✓ | ✓ | **1.22** | **1694** | **0.705** | 32.58 | **0.005** |

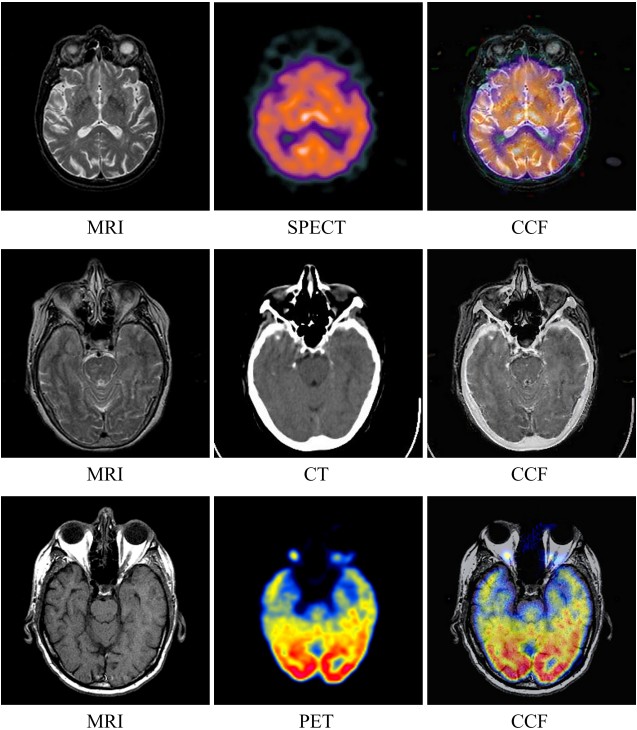

Figure 11: Visualization of medical image fusion.

## H   Experiments on Medical Image Fusion

Medical image fusion (MIF)[59] experiments were conducted to validate the efficacy of our proposed method using the Harvard Medical Image Dataset[60]. We visualized fused medical images, focusing on MRI-SPECT, MRI-CT, and MRI-PET fusion, with the results presented in Fig. 11. Observing our fused images, it is evident that the preservation of both the skull's shape and intensity is notably well-maintained. Within the brain region, the CCF method effectively combines intricate details and structures from the two modalities.

## I   Limitations and Broader Impacts

Even though the proposed CCF model achieves superior performance over existing methods and demonstrates advanced generalization with dynamic condition selection, there are still some potential limitations. We provide a condition bank, but it needs to be constructed empirically. Most of the conditions are inspired by Image Quality Assessment (IQA), but classifying these conditions is challenging due to the complexity of IQA. Therefore, it is necessary to propose a method to automatically classify the conditions, reducing empirical intervention. Additionally, our method relies on a pre-trained diffusion model, which limits its efficiency and makes the generation process time-consuming. Exploring more effective sampling processes would be valuable to improve efficiency and further enhance the model's performance. For potential social impact, it is difficult to ensure the effectiveness of selected conditions for all scenarios, which may be risky in high-risk scenarios such as medical imaging and autonomous driving.

