# OpenReview forum: "Conditional Controllable Image Fusion"
_NeurIPS.cc/2024/Conference — NeurIPS 2024 poster_

### Official Review · Reviewer_ghie · 2024-07-03

**Soundness:** 3
**Presentation:** 2
**Contribution:** 4
**Rating:** 5
**Confidence:** 5

**Summary:**

The paper proposes a controllable conditional image fusion method. This method enables dynamically controllable fusion for each image pairs. The core idea is to empirically construct a conditional bank and dynamically select different control conditions during the diffusion fusion process. The method is suitable for image fusion tasks of different modalities and can also perform controlled fusion given various downstream tasks, such as object detection.

**Strengths:**

- The paper introduces the controllable conditional image fusion algorithm for the first time. This method makes the image fusion process based on the diffusion model controllable through manual feature constraints (empirical).

- This method is suitable for different fusion tasks.

**Weaknesses:**

- The section Sampling-adaptive Condition Selection (SCS) is mainly derived from "GradNorm: Gradient Normalization for Adaptive Loss Balancing in Deep Multitask Networks", which is cited in the paper, but some of the formulas are not clearly expressed. For example, in Eq 11, Gate(C) has C on the left, but the condition C is not contained on the right. Another example is Line 170, Li(t) = E[wi(t)Li(t)]? Additionally, there are some typos, like Line 170, theta.
     Therefore, understanding the routing process without reading the GradNorm paper might be difficult.

- There are some issues with the writing and formatting, such as the order of appearance of Tab1 and Tab2, and the captions for the tables do not specify the task for each tab (e.g., Multi-Focus, Multi-Modal, etc.) in the quantitative comparison results. The captions for the images also have this issue and need improvement.

- There are too many empirical conditions, which may require different settings under various circumstances.

**Questions:**

- In Fig1, the terms “Random”, “Content”, and “Detail” in the center of the diagram need to be clearly explained.

- In Fig2, Top-K Routing is not contained in the main text.

- In Fig3, the enlarged section does not seem to be as good as methods like Swinfusion and Text-IF.

- Line 193, “the feature extracted by a detection network F = D(x)”—what scale/level of feature serves as a good guide? Are there any selection criteria?

- For basic conditions, are the high-frequency information of both modality images (for example,  infrared and visible images) extracted?

- With so many conditions, low-frequency doesn’t seem as important as shown in Fig.1.  Additionally, are there any other empirical conditions with potential?

- In Fig8, the authors mention the network structure, but in Line 493, they state that they used “a pre-trained diffusion model.” So, this network structure should be derived from the paper on the pre-trained model? If so, a citation needs to be provided in the diagram.

**Limitations:**

- The conditional bank is manually designed, and different experiments likely require different fine-tuning. This limits the potential of the proposed method.
- It is necessary to specify the runtime, as time is usually a key metric for diffusion model-based methods.

---

> ### Author Rebuttal · Authors · 2024-08-07
>
> We sincerely thank the reviewer for your valuable comments and appreciate your recognition for our **novelty** on controllable conditional image fusion and **generalization** on different fusion tasks. We believe the constructive feedback will improve the paper and increase its potential impact on the community.
>
> - W1: Formulas and the routing process are not clear.
>
> Eq 11 should be : $Gate=[\omega_1,..\omega_c], \omega_i(t)=\omega_i(t-1)-\bigtriangledown\omega_i$
>
> L170 should be: $L(t)=\mathbb{E}[\omega_i(t)L_i(t)]$
>
> L170 \theta should be $\theta$
>
> **Routing process:**
> (a) **Calculate all enhanced conditions' gradients** using the error of target and $x_0$.
> (b) **Calculate the gradient error** ∇ω with Eq.(12).
> (c) **Update the ω**  acts as '**gate**' in the routing process, with ∇ω.
> (d) **Sort ω** and **identify the corresponding top-k conditions**.
>
> The rooting process facilitates adaptively selecting conditions based on fusion requirements.
>
> - W2: Issues with the writing and formatting.
>
> Thanks for the constructive comment. The MFFW, MEFB and LLVIP datasets refer to Multi-focus, Multi-exposure and Multi-Modal fusion, respectively. We have specified the tasks for each Table.
>
> In addition, we have carefully checked and revised all the presentation issues to mitigate the ambiguity. We appreciate the constructive comments, which improved our presentation.
>
> - W3&L1: Limitations about requiring different settings under various circumstances.
>
> Thanks for the valuable comment. Please kindly note that, for different fusion scenarios, **the condition bank is the same** in our experiments. We propose a Sampling-adaptive Condition Selection (SCS) to **adaptively select different conditions at each denoising step** for different samples, which is sample-adaptive and denoising-adaptive. We have verified the ability of SCS in Sec. 5.5 and App. F, and most metrics show improvement after incorporating SCS.
>
> In addition, we have also conducted experiments on an enlarged condition bank based on related works [5-7], which contains 12 conditions without manually screening. As shown in Tab. A1 of global pdf, while adding more conditions slightly improves performance, it also results in a linear increase in inference time. **To balance the time consumption and performance**, we manually selected 8 conditions as our condition bank.
>
> - Q1:  Need to explain “Random”, “Content”, and  “Detail” in Fig1.
>
> Thanks for the constructive comment. The "Random" refers to the conditions are randomly selected, the "Content" refers to the selection of conditions that content information, and the "Detail" refers to prefer selection of conditions about texture detail.
>
> As discussed in Section 5.6, we examine the preferences for condition selection. We found that image fusion focuses on different conditions at different steps within the diffusion denoising process. The SCS is designed to align the condition selection with the denoising process. Fig.1 demonstrates the effectiveness of our method. Initially, when the image is merely noise without any discernible information, conditions are "**random**" selected. As denoising progresses and content begin to emerge, conditions that aid in synthesizing the '**content**' are chosen. In the later stages, when the focus shifts to refining details, conditions related to '**detail**' are selected.
>
> - Q2: Problem about Top-K Routing.
>
> The Gete is the selection gate, which dynamically chooses the top-k conditions to adaptively generate distinct aspects of the images. As we mentioned the **W1**, the **top-k routing in the SCS for adaptive selection conditions.**
>
> - Q3: Problem about qualitative evaluation of Swinfusion and Text-IF in Fig3.
>
> Thanks for the comment. Compared with SwinFusion, our method outperforms it in the contrast and outline, especially in the manhole cover region producing clearer outlines within the red box.
>
> Compared with Text-IF, our method exhibits higher image definition, particularly evident in the clearer texture details of the broken texture of a zebra crossing within the blue box.
>
> - Q4: The feature selection criteria.
>
> **The deeper feature contains more task-specific information** and can provide a stronger constraint for the task. As shown in Tab. A2 of global pdf, the deepest feature captures the best performance. Additionally, the deepest feature consumes fewer computing resources, making it a better condition.
>
> - Q5: How high-frequency information extracted?
>
> Thanks for the details question. The high-frequency are extracted from both modality images. Then take the max to them for containing more high-frequency information.
>
> - Q6:  Problem about Low-frequency in Fig.1 and potential conditions.
>
> Different conditions have different effects on the fusion results. In our condition bank, content and SSIM can also offer low-frequency constraints, while they also provide other information as well. Since we only select the Top-3 enhanced conditions at each step, the relatively homogeneous low-frequency condition is not selected adaptively. This further verifies our adaptivity, even on a condition bank with redundant conditions.
>
> As mentioned in Question 2, more conditions lead to performance improved with the additional metrics such as CC, MS-SSIM, SCD, and VIFF [5-7] shown in Tab. A1 of global pdf.
>
> - Q7: The citation needs to be provided in Fig. 8.
>
> We visualized the pre-trained DDPM model, as referred to in L95-96 in the manuscript, and we have revised it accordingly.
>
> - L2: Specify the runtime.
>
> As mentioned in **W3&L1**, the runtime is sensitive to the number of conditions, as shown in the Tab. A1 of global pdf. With the addition of conditions, runtime linear increases but the performance improves slightly. So we empirically selected 8 conditions to **balance performance and runtime**. We will emphasize this in the revision.

---

### Official Review · Reviewer_y2va · 2024-07-08

**Soundness:** 3
**Presentation:** 3
**Contribution:** 4
**Rating:** 7
**Confidence:** 4

**Summary:**

This paper proposes a novel Controllable Condition Fusion (CCF) framework that utilizes a pre-trained diffusion model to achieve dynamic and adaptive condition selection without requiring specific training for general image fusion tasks. The authors presented a conditional bank conducted by various conditions to fit diverse scenarios. The conditions can be dynamically and adaptively selected according to the diffusion step, allowing CCF to conditionally calibrate the fused images step by step for each individual sample. Experimental results demonstrate the effectiveness of the proposed method.

**Strengths:**

(1)The paper is well-structured and the proposed idea is novel. It introduces a conditional controllable fusion (CCF) framework for general image fusion tasks without specific training. The generation and denoising capabilities of DDPM are effectively leveraged to produce high-quality fused images, making its integration with image fusion both ingenious and well-suited.
(2)A new dynamic conditional paradigm was introduced with a conditional bank that regulates image fusion. This allows for the adaptive selection of multiple conditions, enabling control over various image fusion processes.
(3)The paper validates the superior performance of the CCF framework over SOTA methods through extensive experiments on various fusion tasks. This enhances the credibility and applicability of the proposed approach.

**Weaknesses:**

(1)This paper introduces a controllable condition image fusion model. However, this ‘condition’ is not the common guidance condition, such as sketch, location map, pose image, etc, but evaluation metrics, such as SSIM, Edge Intensity et al. Therefore, suggesting that authors give a clear explanation or definition of the condition in Section Introduction.
(2)An important comparison is missing. From the first impression, the proposed method is to leverage the reconstruction capability of DDPM. However, the article did not demonstrate that the fusion process utilized the reconstruction capability of DDPM, it is important to prove it systematically.
(3)One of the core innovations in this paper is the Conditional Bank, which contains three types of conditions: basic fusion conditions, enhanced fusion conditions, and task-specific fusion conditions. However, there lacks a clear definition of three conditions to identify the difference between these conditions.
(4)The paper contains some small mistakes in symbol, such as Line 170 theta, the lack of a subscript for eq 6 \epsilon and missing the parentheses. Additionally, there is indistinct use of "x0|t" and "x0" in Line 134. Overall, the work is very readable, but the problems with the writing should be corrected.

**Questions:**

Please refer to the weaknesses above.

**Limitations:**

The authors have discussed the limitations of this work. And I don’t think there are any direct negative societal implications. Other limitations and opportunities for improvement are addressed in my responses to previous questions.

---

> ### Author Rebuttal · Authors · 2024-08-07
>
> We'd like to thank the reviewer for the valuable comments, and acknowledgment of our **novel**, **ingenious integration** and over current **state-of-the-art** framework. We appreciate your support and constructive suggestions and address your concerns as follows.
>
> - W1: This paper introduces a controllable condition image fusion model. However, this ‘condition’ is not the common guidance condition, such as sketch, location map, pose image, etc, but evaluation metrics, such as SSIM, Edge Intensity et al. Therefore, suggesting that authors give a clear explanation or definition of the condition in Section Introduction.
>
> It would kindly note that the condition is not similar to stable diffusion, it is the fusion condition in this paper. **The fusion condition aims to synthesize a unified image that amalgamates complementary information from multiple images**. For typical fixed conditions scenarios, we often design conditions to determine which information needs to be fused such as $||x_0,V||_2$ and $||x_0,I||_2$ . Therefore, we propose the conditional bank with Sampling-adaptive Condition Selection (SCS) for selecting conditions to adapt different task scenarios.
>
> - W2: An important comparison is missing. From the first impression, the proposed method is to leverage the reconstruction capability of DDPM. However, the article did not demonstrate that the fusion process utilized the reconstruction capability of DDPM, it is important to prove it systematically.
>
> Thank you for your detailed comment. The refinement of the denoising process in DDPM is the denoising process that removes the noise progressively, step by step, and produces clearer and more realistic images. Unlike other methods with fixed fusion conditions, the CCF framework **dynamically selects the fusion conditions from a condition bank**. DDPM decomposes image fusion across multiple steps, enabling combined CCF so that conditions can be injected iteratively during the denoising process to guide the generation toward the final fused images. We have found that image fusion focuses on different conditions at different steps within the diffusion denoising process. This nuanced understanding inspired the proposal of SCS, which **adaptively selects the suitable condition at each sampling step**, aligning with the generation preferences to optimize image fusion. As illustrated in the Tab. A3 of global pdf shows significant improvement with the reconstruction capability of DDPM.
>
>
> - W3: One of the core innovations in this paper is the Conditional Bank, which contains three types of conditions: basic fusion conditions, enhanced fusion conditions, and task-specific fusion conditions. However, it lacks a clear definition of three conditions to identify the difference between these conditions.
>
> Thanks for the valuable comment. We establish a condition bank that includes basic, enhanced, and task-specific fusion conditions. The basic conditions are **essential for obtaining the fundamental** fused image, while the enhanced conditions are adaptively selected using SCS to **enhance the quality** of the synthesized fused image. Additionally, task-specific conditions can be manually selected to **obtain more task perceptions** in the fused images.
>
>
> - W4: The paper contains some small mistakes in symbols, such as Line 170 theta, the lack of a subscript for eq 6 \epsilon and missing the parentheses. Additionally, there is indistinct use of "x0|t" and "x0" in Line 134. Overall, the work is very readable, but the problems with the writing should be corrected.
>
> L170 should be $\theta$
>
> $x_{0|t} \approx f_\theta(x_t, t) = \frac{(x_t -\sqrt{1-\bar{\alpha_t}}\epsilon_\theta(x_t, t))}{\sqrt{\bar{\alpha}_t}}$
>
> We have carefully checked and revised all the presentation issues throughout the paper. We appreciate the reviewer's constructive comments, which greatly improved our presentation.

---

> > ### Comment · Reviewer_y2va · 2024-08-11
> >
> > Thanks for your responses. My concerns have been well solved. Based on the novel design of conditional controllable fusion framework and sufficient experiments, I tend to increase my score to 7.

---

> > > ### Author Response · Authors · 2024-08-11
> > > **Great thanks for the positive reply!**
> > >
> > > We sincerely thank the reviewer for the positive reply. Meanwhile, we are very grateful that the reviewer recognizes our responses and work, while increasing the final rating to 'accept'. Great thanks again!

---

### Official Review · Reviewer_PorP · 2024-07-10

**Soundness:** 3
**Presentation:** 2
**Contribution:** 2
**Rating:** 5
**Confidence:** 5

**Summary:**

This paper proposes a diffusion-based image fusion method with adaptive fusion conditions. It aims to solve the drawback of existing method, i.e., the application of distinct constraint designs tailored to specific scenes. This method builds a condition bank with basic, enhanced, and task-specific conditions. Then, it employs specific fusion constraints based on these conditions for each individual in practice. The proposed method is tested on various image fusion tasks, including multi-modal, multi-exposure, and multi-focus image fusion.

**Strengths:**

1. The proposed method can combine the advantages of multiple types of loss functions and take consideration of downstream tasks, thus dynamically adjust the optimization direction during the sampling process.
2. This method is applicable to multiple image fusion tasks, including visible-infrared, medical, multi-exposure, and multi-focus image fusion.
3. The experiments are rich and the results are competitive.

**Weaknesses:**

1. The basic and enhanced conditions are some widely used image fusion constraints, and the task-specific condition is based on the feature extracted by a downstream network. Thus, the main contribution lies in the design of the gate of conditions. However, the reason why is the gate defined in this way and the principle behind this definition not fully explained in detail.
2. It focuses on the problem that such data-driven fusion methods are hardly applicable to all scenarios, especially in rapidly changing environments and source images with dynamic differences. However, the experiment fails to show the advantages in these scenarios.
3. This method states that multiple conditions can be considered, but the actual considerations are limited, such as the SSIM-based enhanced condition and detection-based task-specific condition.
4. The article basically describes the process of the method, but the motivation, principle, and details of the method are not clear and explicit enough.

**Questions:**

1. What does $s_\theta(x_t,t)$ in Eq. (10) represent? And what is the form of $C$ in Eq. (10)? As mentioned in the multiple types of conditions, some conditions are in pixel level, such as MSE, SSIM, feature similarity, while some is denoted with the characteristic, such as SD. The form is not unified, and the characteristic-based condition may have some effect on the content.
2. In the specific conditions, how to build the constraint between F(x_0) and F(V,I), as the first term takes a single image as input while the last term take the two source images as the input.
3. For the task-specific condition, the extracted features with the task network is used for constraint. Why not use the final performance of the task for constraint?

**Limitations:**

1. The first and second contributions are essentially the same.
2. The conditions discussed in the Appendix are almost different basic conditions. And the results of MSE are a bit strange.
3. It lacks the experiment on challenging and dynamic scenarios.

---

> ### Author Rebuttal · Authors · 2024-08-07
>
> We would like to thank the reviewer for recognizing our method as **applicable to multiple fusion tasks, rich experiments** and **competitive results**. We will also make an effort to increase clarity throughout.
>
> - W1&L1: More explanations for contributions and the "gate" of conditions.
>
> Thanks for the constructive comments.
>
> i) The first contribution focuses on the **dynamically controllable fusion framework**, which integrates various fusion constraints as a condition bank. The condition bank contains basic condition (BC), enhanced condition (EC), and task-specific condition (TC). Thus, we can **select different conditions to perform fusion for different samples**. Please kindly note that this is a sample-level selection for the whole denoising process.
>
> ii) Our second contribution lies in the proposed SCS, which adaptively injects suitable conditions at each denoising step of diffusion model. To our knowledge, **we first found that the diffusion-based image fusion model requires significantly different constraints at different steps.** This inspired us to perform **different selections for each step of denoising step**, endowing more adaptivity.
>
> We have revised the contributions as:
>
> - We propose a pioneering conditional controllable image fusion (CCF) framework with a condition bank, achieving controllability in various image fusion scenarios and facilitating the capability of dynamic controllable image fusion.
>
> - We present the Sampling-adaptive Condition Selection (SCS) to subtly integrate the condition bank into denoising steps of diffusion, allowing adaptively selected conditions on the fly without additional training and ensuring the dynamic adaptability of the fusion process.
>
> iii) The "gate" refers to the routing process:
> (a) **Calculate all enhanced conditions' gradients** using the error of target and $x_0$. (b) **Calculate the gradient error** ∇ω with Eq.(12). (c) **Update the ω**  acts as '**gate**' in the routing process, with ∇ω. (d) **Sort ω** and **identify the corresponding top-k conditions**. The routing process facilitates the adaptive selection of conditions based on fusion requirements.
>
> - W2&L3: Experiments on challenging scenarios.
>
> Thanks for the valuable comment. We have added experiments on rapidly changing scenarios. Shows in Fig. A1 of global pdf, our CCF adapts to changing environments by selecting different conditions at different sampling steps.
>
> - W3: Limitations about actual consideration of condition.
>
> i) The EC and TC are **not limited to SSIM and detection**. The ECs include 8 conditions (L490-491), and TCs are not limited to object detection, classification, segmentation, and depth, as shown in Fig. 2.
> ii) The ECs are **adaptively selected at each denoising step by SCS** module, and the **condition bank is expandable** for more potential tasks. Tab. A1 shows that more ECs improve the performance.
>
> - W4: Motivation, principle, and details.
>
> **Motivation:** The dynamically changing environments lead to varying effective information of different modalities, which is sensitive to different constraints. This inspired us to assign different constraints as the conditions to improve flexibility of fusion. Our CCF subtly introduces a condition bank to diffusion model, injecting suitable conditions at each denoising step according to diffierent samples.
>
> **Principle:** We first found the diffusion-based image fusion model prefers significantly different constraints at different denoising steps (see Fig. 1). Based on this, we proposed SCS to adaptively choose suitable conditions at each step, progressively optimizing the fusion results.
>
> **Details:** Our method is based on a pre-trained DDPM. During the denoising process, we use conditions to constrain $x_0$ for image fusion. The BCs ensure fundamental fusion across various fusion tasks, while the ECs are adaptively selected by SCS (L145-151) to improve the fused image at each step. TCs is manually selected to introduce more task-specific constraints.
>
> - Q1: Presentation issues.
>
> $s_\theta(x_t, t)$ and $f_{\theta}$ all refers to the estimation of $x_0$ at t-th step. We would unify $s_\theta(x_t, t)$ into $f_{\theta}$.
>
> C is the target of the conditions. Take the MSE as an example, the  $||C-\mathcal{A}(x_0)||_2$ can be express as $||y- x_0||_2$
>
> The conditions are not classified by pixel level or characteristic. The classification criteria of different conditions are based on the selection frequency. The content condition refers to the basic condition in our paper. To avoid ambiguity, we will replace 'content' with 'basic' in the revision.
>
> - Q2: The constraint of TCs.
>
> Thanks for the valuable question. To mitigate the ambiguity, we have revised it to $||F(X), F(M)||_2$ where $X\in \{x_0\}_i^m$ and $M$ is the set of $m$ modalities.
>
> - Q3: The final performance of the task for constraint?
>
> In general, the task-specific ground truth is not available. Therefore, the final performance of the task is not used for constraint. Instead, we introduce the feature and final results of task network on source images as the pseudo label to constrain the denoising process, integrating task-specific information for more robust fusion.
>
> In addition, we have added experiments to compare the effect of feature constraint before and after the detection neck. Tab. A2 in global pdf indicates deep features before detection neck showing superior performance as they balance the detection-specific information and sufficient representation.
>
>
> - L2: More explanations for basic conditions and MSE.
>
> Considering the significant gaps between different tasks, we propose a set of BCs tailored various task scenarios. E.g., in the MMF task, MSE [3-6] is a commonly used condition. In both MMF and MEF tasks, to achieve clearer and higher fidelity images, Wavelets are widely employed [8-11].
>
> The MSE metric [12] may be different due to the image size, value range (e.g., 0-1 or 0-255), and channel configuration (e.g., RGB or grayscale).

---

> ### Author Response · Authors · 2024-08-13
>
> Dear Reviewer PorP,
>
> We sincerely appreciate the time and effort you have dedicated to reviewing our submission, and we thank a lot for your constructive comments on our paper. Given that the discussion period is around the corner, would you please kindly let us know if our response has addressed your concerns? If you have any further questions, please let us know. We would be more than happy to clarify more about our paper and discuss it further with you.
>
> Best regards,
>
> Authors of Submission 35

---

> > ### Comment · Reviewer_PorP · 2024-08-13
> >
> > Thank you for the rebuttal. After reviewing the other reviewers' comments and your response, some of my concerns and questions have been addressed. I lean to increasing my final score to 5.

---

> ### Author Response · Authors · 2024-08-13
>
> Thank you for your positive feedback! We are very grateful that the reviewer recognizes our responses and work. Your insightful comments have greatly improved our work and inspired us to research more. If you have any further questions, please let us know. We would be more than happy to clarify more about our paper and discuss it further with you.

---

### Official Review · Reviewer_spMU · 2024-07-17

**Soundness:** 3
**Presentation:** 2
**Contribution:** 2
**Rating:** 6
**Confidence:** 5

**Summary:**

This paper proposes a Conditional Controllable Fusion framework called CCF, effectively addressing the issue that existing data-driven fusion methods struggle to adapt to all scenarios. The authors conducted extensive experiments to demonstrate the effectiveness of the CCF. This manuscript is standardized and the writing is fluent.

**Strengths:**

1. The idea of conditional controllable image fusion (CCF) is novel.
2. Extensive experiments are conducted to demonstrate the effectiveness of the CCF.
3. This manuscript adheres to a high writing standard, ensuring fluent and easily understandable content.

**Weaknesses:**

1. How was the Selection frequency map in Figure 1 obtained? Is it based on real data or simulated?
2. It is not clear exactly what iterative refinement of sampling refers to. Why does sampling need to be iteratively refined? The sampling iteration is derived from the diffusion model DDPM and how is it related to the proposed CCF?
3. What are the advantages of using the DDPM for controlled image fusion? In other words, how do diffusion models contribute to the adaptability and controllability of image fusion?
4. Multiple sign ambiguities. Including but not limited to: 1) $c$ denotes both the channels and the given condition in Eq. (5). 2) What does the $f_{\theta}$ refer to? 3) What is the $s_{\theta}$?
5. How are task-specific conditions incorporated into DDPM? Lack of specific implementation details.
6. The theoretical discussion of adaptive customization of conditions for each sample is limited to sections 4.1 and 4.2, lacking specific customization procedures or visual case studies. Consequently, the credibility of this contribution in terms of condition customization is compromised.
7. The authors chose 8 enhanced conditions, i.e., SSIM, Content, Edge, Low-frequency, High-frequency, Spatial Frequency, Edge Intensity, and Standard Deviation enhancements. However, their selection lacks a strong foundation or rationale.

**Questions:**

1. Could you kindly elucidate the methodology employed to generate the Selection frequency map depicted in Figure 1? Real or simulated?
2. Why Use DDPM to achieve conditional image fusion?
3. Can you give an example to explain the specific process of condition selection in the process of sampling iteration?

**Limitations:**

Please refer to "Weakness".

---

> ### Author Rebuttal · Authors · 2024-08-07
>
> We'd like to thank the reviewer for the valuable comments and appreciate your recognition of the **novel idea**, **effective method** and **fluent writing**. We provide detailed responses to the constructive comments.
>
> - W1&Q1: How was the Selection frequency map in Figure 1 obtained? Is it based on real data or simulated?
>
> Thanks for your constructive comment. The **statistical result** in Fig. 1 is based on the **real data**. For each sample, the conditions are selected by the Sampling-adaptive Condition Selection (SCS) (L159, Sec. 4.2 ) from the enhanced conditions at each denoising step of diffusion. We counted the frequency of all sample selection conditions in LLVIP. The deeper the color, the higher the frequency.
>
> - W2&Q2:  More explanations for sampling iteration.
>
>
> Thanks for your detailed comment. The denoising process is also named as the inverse diffusion process or sampling process [1]. The refinement of sampling in DDPM is the denoising process that removes the noise progressively and produces more realistic images. Please kindly note that we for the first time found the diffusion-based image fusion model requires significant different conditions at different steps. This inspired us to **subtly integrate** **DDPM** into our dynamic fusion framework and proposed SCS to **adaptively select the suitable conditions** at each sampling step. Specifically, we inject the **selected conditions iteratively** during the sampling process to guide the generation and refine the fused images. As illustrated in Fig. 1, it proved our condition selection is consistent with the generation of DDPM.
>
> - W3: How do diffusion models contribute to the adaptability and controllability of image fusion?
>
>
> Thanks for your detailed comment.
>
> i) As previously mentioned in **W2&Q2**, DDPM generates images progressively, with conditions being injected at each diffusion denoising step. We decompose the image fusion process into several controllable parts, allowing each part to be controlled individually, thereby **ensuring overall controllability**.
>
> ii) In the process of progressively fusing images with diffusion, different steps have varying perceptions of the images, as shown in Fig 1. As discussed in Sec. 5.6, in the initial stage, the conditions are randomly selected, the middle stage prefers conditions related to content, and the final stage focuses on conditions related to the details. Therefore, our CCF is qualified to **adaptively select suitable conditions at different stages** of the denoising process to effectively fuse the image with DDPM.
>
>
> - W4:  Multiple sign ambiguities. Including but not limited to: 1) c denotes both the channels and the given condition in Eq. (5). 2) What does the $f_\theta$ refer to? 3) What is the $s_\theta$?
>
>
> Thanks for the valuable suggestions.
>
> i) Eq. (5) $c$ represents the given condition. To eliminate ambiguity, we have revised the $n$ signifies the channels in L123-124.
>
> ii) $f_{\theta}$ and $s_\theta$ both denotes the estimation of $x_0$. To eliminate ambiguity, we have unified them as $f_{\theta}$.
>
> We have conducted a thorough review to eliminate all ambiguous notations and to clarify the expressions.
>
>
> - W5:  How are task-specific conditions incorporated into DDPM? Lack of specific implementation details.
>
>
> Thanks for the constructive comment. As described in **L189-196**, the task-specific conditions are **incorporated to constrain the fusion across the whole denoising process**. For instance, we take the Euclidean distance of features extracted by object detection model as the detection condition. We deploy YOLOv5 in the experiments, which extracts the features of estimated $x_0$ in each step and visible image, as the YOLOv5 is pretrained on visible modality. We minimize the Euclidean distance in the inverse diffusion process iteratively. Consequently, the final fused image progressively integrates the object-specific information, enhancing the fusion performance.
>
>
> - W6&Q3:  More explanations for the specific process of condition selection in the process of sampling iteration?
>
>
> As shown in Fig. 1, we analyze the condition selection at each step for all samples in the LLVIP dataset. Furthermore we explain the process of condition selection in detail.
>
> For example, in the MMF task in our experimental setting, using the visible image $v$ and infrared image $i$.
>
> 1. **Build the conditional bank**: Detailed in Appendix A, "Experimental Settings."
> 2. **Estimate $x_0$** with Eq.(6).
> 3. **Calculate each condition** using the $v$ and $i$.
> 4. **Obtain** $\omega$ with Eq.(11) .
> 5. **Select the index of conditions** same as $\text{topk}(\omega)$ for image fusion.
> 6. **Loop to the end** of the denoising process and finally **get the fused image**.
>
> We have extended the visual case studies of the sampling process iterations in Fig. A1 of global pdf.
>
> - W7:  The authors chose 8 enhanced conditions, i.e., SSIM, Content, Edge, Low-frequency, High-frequency, Spatial Frequency, Edge Intensity, and Standard Deviation enhancements. However, their selection lacks a strong foundation or rationale
>
> As shown in **L181-188**, we follow the recent works [2-4] and **choose 8 most used constraints as the conditions**. However, we are not limited to these, as shown Tab. A1 of global pdf, **more enhanced conditions (CC, MS-SSIM, SCD, VIFF)[5-7] can be incorporated** . While adding more conditions slightly improves performance, it also results in a linear increase in inference runtime.

---

> > ### Comment · Reviewer_spMU · 2024-08-13
> > **Thank you for the detailed response.**
> >
> > Thank you for the detailed response. My concerns have been well addressed. Accordingly, I have raised my rating and highly recommend the authors include those discussions on adaptive conditions in the revised version.

---

> > > ### Author Response · Authors · 2024-08-13
> > >
> > > Great thanks for your support! We sincerely appreciate your constructive comments and will carefully revise our paper accordingly.

---

### Author Rebuttal · Authors · 2024-08-07

Dear PCs, SACs, ACs, and Reviewers,



We would like to thank you for your valuable feedback and insightful reviews, which have greatly contributed to improving the paper. This is a **fluent** and **well-structured** (Reviewer spMU, y2va) manuscript with a **novel** idea (Reviewer spMU, y2va), we proposed a **new** Controllable Condition Fusion (CCF) framework (Reviewer y2va, ghie), **extensive** experiments (Reviewer spMU, PorP, y2va) on multiple tasks validate CCF’s results are **competitive** (Reviewer PorP) and **effectiveness** (Reviewer spMU, y2va, ghie) achieving **SOTA** (Reviewer y2va). Our framework is **suitable across different image fusion scenarios** (Reviewer spMU, PorP, y2va, ghie).

We hope that our responses will satisfactorily address your questions and concerns. We sincerely appreciate the time and effort you have dedicated to reviewing our submission, along with your invaluable suggestions. All the missing details will be added in the revision, and we will also release all our codes to ensure clarity and reproducibility.



Sincerely,

Authors



## Reference

[1] Ho J, Jain A, Abbeel P. Denoising diffusion probabilistic models[J]. Advances in neural information processing systems, 2020, 33: 6840-6851.

[2] Zhao Z, Bai H, Zhang J, et al. Cddfuse: Correlation-driven dual-branch feature decomposition for multi-modality image fusion[C]//Proceedings of the IEEE/CVF conference on computer vision and pattern recognition. 2023: 5906-5916.

[3] Zhao Z, Bai H, Zhu Y, et al. DDFM: denoising diffusion model for multi-modality image fusion[C]//Proceedings of the IEEE/CVF International Conference on Computer Vision. 2023: 8082-8093.

[4] Cheng C, Xu T, Wu X J. MUFusion: A general unsupervised image fusion network based on memory unit[J]. Information Fusion, 2023, 92: 80-92.

[5] Xu H, Ma J, Jiang J, et al. U2Fusion: A unified unsupervised image fusion network[J]. IEEE Transactions on Pattern Analysis and Machine Intelligence, 2020, 44(1): 502-518.

[6] Zhu P, Sun Y, Cao B, et al. Task-customized mixture of adapters for general image fusion[C]//Proceedings of the IEEE/CVF Conference on Computer Vision and Pattern Recognition. 2024: 7099-7108.

[7] Li H, Wu X J, Kittler J. RFN-Nest: An end-to-end residual fusion network for infrared and visible images[J]. Information Fusion, 2021, 73: 72-86.

[8] Ma J, Chen C, Li C, et al. Infrared and visible image fusion via gradient transfer and total variation minimization[J]. Information Fusion, 2016, 31: 100-109.

[9] Liu Y, Wang L, Cheng J, et al. Multi-focus image fusion: A survey of the state of the art[J]. Information Fusion, 2020, 64: 71-91.

[10] Zhang W, Liu X, Wang W, et al. Multi-exposure image fusion based on wavelet transform[J]. International Journal of Advanced Robotic Systems, 2018, 15(2): 1729881418768939.

[11] Zhang J, Wu M, Cao W, et al. Partition-Based Image Exposure Correction via Wavelet-Based High Frequency Restoration[C]//International Conference on Intelligent Computing. Singapore: Springer Nature Singapore, 2024: 452-463.

[12] Xu Y, Li X, Jie Y, et al. Simultaneous Tri-Modal Medical Image Fusion and Super-Resolution using Conditional Diffusion Model[J]. arXiv preprint arXiv:2404.17357, 2024.

---

### Decision · Program_Chairs · 2024-09-25

**Decision:**

Accept (poster)

**Comment:**

The paper proposes a method that enables dynamic conditional controllable fusion for image pairs with reported excellent performance.

The authors provided satisfactory responses and convinced
- Reviewer spMU to increase the score to Weak Accept
- Reviewer PorP to increase the score to Borderline Accept
- Reviewer y2va to increase the score to Accept
- while Reviewer ghie maintains Borderline Accept.

Thus, for this work there is an acceptance consensus.

After reading the paper and carefully checking the reviews and the authors' responses the ACs agree with the reviews that the present work makes important contributions and is of interest for the community.

The authors are invited to further refine their paper for the camera ready by including (part of) information/details from their responses to the reviewers' comments.